# Aberrant APOBEC3B Expression in Breast Cancer Is Linked to Proliferation and Cell Cycle Phase

**DOI:** 10.3390/cells12081185

**Published:** 2023-04-18

**Authors:** Pieter A. Roelofs, Mieke A. M. Timmermans, Bojana Stefanovska, Myrthe A. den Boestert, Amber W. M. van den Borne, Hayri E. Balcioglu, Anita M. Trapman, Reuben S. Harris, John W. M. Martens, Paul N. Span

**Affiliations:** 1Radiotherapy & OncoImmunology Laboratory, Department of Radiation Oncology, Radboud University Medical Center, 6525 GA Nijmegen, The Netherlands; 2Department of Biochemistry, Molecular Biology and Biophysics, Masonic Cancer Center, Institute for Molecular Virology, and Center for Genome Engineering, University of Minnesota, Minneapolis, MN 55455, USA; 3Department of Medical Oncology, Erasmus MC Cancer Institute, Erasmus University Medical Center, 3000 CA Rotterdam, The Netherlands; 4Department of Biochemistry and Structural Biology, University of Texas Health San Antonio, San Antonio, TX 78229, USA; 5Howard Hughes Medical Institute, University of Texas Health San Antonio, San Antonio, TX 78229, USA

**Keywords:** APOBEC3B, cell cycle, transcription, G_2_/M, RB/E2F pathway, PKC/ncNF-κB pathway

## Abstract

APOBEC3B (A3B) is aberrantly overexpressed in a subset of breast cancers, where it associates with advanced disease, poor prognosis, and treatment resistance, yet the causes of A3B dysregulation in breast cancer remain unclear. Here, A3B mRNA and protein expression levels were quantified in different cell lines and breast tumors and related to cell cycle markers using RT-qPCR and multiplex immunofluorescence imaging. The inducibility of A3B expression during the cell cycle was additionally addressed after cell cycle synchronization with multiple methods. First, we found that A3B protein levels within cell lines and tumors are heterogeneous and associate strongly with the proliferation marker Cyclin B1 characteristic of the G_2_/M phase of the cell cycle. Second, in multiple breast cancer cell lines with high *A3B*, expression levels were observed to oscillate throughout the cell cycle and again associate with Cyclin B1. Third, induction of *A3B* expression is potently repressed throughout G_0_/early G_1_, likely by RB/E2F pathway effector proteins. Fourth, in cells with low A3B, induction of A3B through the PKC/ncNF-κB pathway occurs predominantly in actively proliferating cells and is largely absent in cells arrested in G_0_. Altogether, these results support a model in which dysregulated A3B overexpression in breast cancer is the cumulative result of proliferation-associated relief from repression with concomitant pathway activation during the G_2_/M phase of the cell cycle.

## 1. Introduction

Normal and cancerous mammary epithelial cells are subject to a multitude of endogenous and exogenous sources of mutagenesis, scarring their genome with a barrage of genetic alterations [1,2,3]. Well-known sources that inflict these mutations are reactive oxygen species, inadequate repair of DNA lesions due to repair deficiencies, and carcinogenic compounds present in tobacco smoke [3]. Additionally, the mutational landscape of certain cancer subtypes, including breast cancer, is drastically shaped by members of the APOBEC family of deaminases, some of which catalyze the deamination of cytosines in genomic single-stranded DNA substrates [4,5,6,7,8]. Mutagenesis mediated by these DNA cytosine deaminases, referred to as APOBEC mutagenesis, actively contributes to inter- and intra-tumoral heterogeneity and increases in strength during the progression of breast cancer [9,10,11], as reviewed in [12]. In fact, together with spontaneous deamination of 5-methylcytosines and homologous recombination deficiency, APOBEC mutagenesis is one of the main sources of genomic scarring in breast cancer genomes and is a major mechanism of endogenous mutations [3,8,9,11]. For instance, mutations in known drivers of breast cancer such as *PIK3CA*, *KMT2C*, *ARID1A*, *NF1*, and *CDH1* have been attributed to APOBEC enzymatic activity and are enriched in metastatic breast cancer, supporting a model in which APOBEC proteins are promoting tumor evolution [9,11]. The activity of this family of enzymes also contributes to poor outcomes in response to breast cancer therapies [13,14] and is reviewed in [12].

Humans express up to seven different APOBEC3 (A3) enzymes, namely A3A, A3B, A3C, A3D, A3F, A3G, and A3H. To date, the most likely family members to contribute to APOBEC mutagenesis are A3A and A3B, with abundant evidence supporting a role for A3B in breast cancer [5,13,15,16,17,18], as reviewed in [19]. *A3B*, which is expressed at a low level in normal mammary tissue, is overexpressed in a subset of cancerous mammary tissues [5,15,20], as reviewed in [19]. In addition, high *A3B* expression in breast cancer is associated with aggressive clinicopathological characteristics [15,16,17,18] resulting in increased metastatic potential, as well as worse prognosis in treatment-naïve disease [13,16,17]. Therefore, considering the significance of A3B in breast cancer, determining what drives this protein to abnormally high expression levels is critically important. 

Earlier work by our group and others has pointed out that both A3B mRNA and protein levels associate with proliferation markers in multiple tumor types [21,22,23,24,25,26,27,28]. Moreover, negative regulation of *A3B* transcription is facilitated through two distinct repressive E2F complexes, at least one of which associates with the cell cycle [24,25,29]. However, it remains unknown whether aberrant A3B expression in breast cancer is caused by de-repression and/or enhanced induction. We therefore set out to investigate the proliferation-dependency of A3B expression and induction throughout the cell cycle of breast cancer cells. The findings presented here support a model in which A3B expression is universally repressed during cellular dormancy (i.e., G_0_ and G_1_ phases), whereas pathway activation during the proliferative stage (G_2_ and M phases) contributes to the overexpression of A3B found in a subset of breast cancer cell lines and tumors. Together, these results provide a molecular explanation for the association of A3B expression and cancer cell proliferation and provide further insights into the transcriptional (dys)regulation of this cancer genomic DNA deaminase.

## 2. Materials and Methods

### 2.1. Cell Lines and Culture Conditions

All cell lines used in this study were cultured at 37 °C under 5% CO_2_. MCF10A cells and the derived HDR and control clones were previously described and cultured using an identical medium recipe [24]. MCF7 cells were cultured in DMEM (Gibco) supplemented with 10% FCS, Glutamax (Thermo Fisher Scientific, Waltham, MA, USA), 6 µg/L recombinant human insulin (Sigma Aldrich, St. Louis, MO, USA), penicillin (100 U/mL), and streptomycin (100 µg/mL). HCC1599, HCC1954, SUM225CWN, and HCC1143 cells were cultured in RPMI supplemented with 10% FCS, penicillin (100 U/mL), and streptomycin (100 µg/mL). SKBR3, MDA-MB-415, BT474, and MDA-MB-468 cells were cultured in DMEM supplemented with 10% FCS, Glutamax, penicillin (100 U/mL), and streptomycin (100 µg/mL). All cell lines routinely tested negative for *Mycoplasma* using the MycoAlert Mycoplasma Detection Kit (Lonza, Basel, Switzerland). All cell counting took place using automated cell counters (Countess, Thermofisher and Luna, Logos Biosystems). PMA was always used at 25 ng/mL for 6 h. Knockdown of *A3B* was performed using shRNA technology, as described in [5]. An overview of the most relevant cell lines and their characteristics relating to this study is provided in Appendix A.

### 2.2. Tissue Collection and (Multiplex) IHC

Cell lines used in conventional and multiplex IHC assays were part of an in-house cell line microarray (CMA), which contains 54 formalin-fixed paraffin-embedded (FFPE) breast cancer cell lines. Primary breast cancer specimens were obtained under the protocol to study biological markers associated with disease outcome and in accordance with the Code of Conduct of the Federation of Medical Scientific Societies in the Netherlands (https://www.coreon.org/wp-content/uploads/2020/04/coreon-code-of-conduct-english.pdf, accessed on 14 April 2023) and the new European General Data Protection Regulation (GDPR). This protocol states that the use of coded left-over material for scientific purposes and, therefore, for the greater good, does not require informed consent according to Dutch law. This waiver for informed consent was acknowledged by the medical ethics committee of the Erasmus Medical Centre Rotterdam, the Netherlands, in MEC 02.953.

Detection of A3B by conventional IHC on these cell lines was performed as described in [26,27,28]. For multiplex IHC, slides containing cell lines or primary breast cancer specimens were rehydrated by three xylene submersions of 10 min, followed by three 100% ethanol washes, one 70% ethanol wash, one 50% ethanol wash, and a final wash in ddH_2_O. Tissue material was then fixed to the glass slides by a 10-minute incubation in neutral buffered formalin, followed by a wash in ddH_2_O. For antigen retrieval, a 15-minute heating cycle at 95 °C in 1X AR6 buffer (Akoya Biosciences, AR600250) was performed. Subsequently, multiplex staining was performed for A3B, Cyclin B1, Cyclin E2, and pan-Cytokeratin. In short, non-specific epitopes were blocked in blocking solution (Akoya Biosciences Marlborough, MA, USA, ARD 1001EA) for 10 min at RT, and rabbit anti-A3B (5210-87-13 [26]) was added at a 1:640 dilution in blocking solution. After a combined incubation of 1 h at RT and 4 °C O/N, slides were washed thrice in PBS-T, followed by another 10-minute blocking step. Opal HRP Polymer Ms + Rb (Akoya Biosciences, ARH 1001EA) was added for 10 min, followed by PBS-T wash. Opal 650 (Akoya Biosciences, FP1496001KT) was added at a 1:200 dilution in amplification diluent (Akoya Biosciences, FP1498) and incubated for 10 min at RT, followed by three PBS-T washes. This procedure was then repeated using rabbit anti-Cyclin B1, 1:25 (Santa Cruz, Dallas, TX, USA, clone GNS1, sc-247, 1 h RT + 4 °C O/N), using the Opal 520 fluorophore, 1:200 (Akoya Biosciences, FP1487001KT); mouse anti-Cyclin E2, 1:100 (Santa Cruz, sc-245, 1 h RT), using the Opal 690 fluorophore, 1:200 (Akoya Biosciences, FP1497001KT); and mouse anti-pan-Cytokeratin (pan CK), 1:400 (Novus Biologicals, Englewood, CO, USA, clone AE-1/AE-3, 30 min RT), using the Opal 620 fluorophore, 1:200 (Akoya Biosciences, FP1495001KT). The slides were mounted after DAPI counterstaining.

Spectral library mapping was performed using single-stained sample slides of BT474 cells using the Vectra 3.0 system. This library was used for spectral overlap compensation and to determine the set exposure time per filter channel of the Vectra microscope. Grey-scale images of samples were taken using inform advanced image analysis software package (PerkinElmer) and exported to a TME-Analyzer version 2.4 beta (Balcioglu et al., manuscript in preparation), which was utilized to segment cells based on cytokeratin (i.e., negative, borderline, or positive) staining and appearance (i.e., grouped, individual, or spindle-shaped cells). A Python code was written (available upon request) to process data provided by the TME-Analyzer and determine co-localization at a cellular level. To determine the likelihood of the found co-localization occurring naturally, binomial distribution was performed using the following Formula (1):(1)$$P_{\mathrm{bin}}=\left(\begin{array}{c} n\\ k \end{array}\right)p^{k}{\left(1-p\right)}^{n-k}$$
where:n = the total cells per cell line;k = the number of double positives (i.e., a cell staining positive for A3B and Cyclin B1/Cyclin E2);p = the chance of a double-positive cell based on the occurrence of the individual Cyclin per cell line, as calculated using the following Formula (2):
(2)$$\left(\frac{\#A3B}{\#\mathrm{Total}}\cdot \frac{\#\mathrm{Cyclin}}{\#\mathrm{Total}}\right)$$

The null hypothesis of the binomial test stated that the amount of double-positive cells (e.g., A3B and Cyclin B1) was caused by chance, and the alternative hypothesis stated that co-localization was caused by a yet-to-be-determined underlying biological mechanism. The chosen *p*-value of 0.05 was corrected with a Bonferroni correction. The null hypothesis was rejected when *p*-values were lower than 0.05.

### 2.3. Cell Cycle Synchronization

MCF10A cells were synchronized in G_0_ by a combined method involving contact inhibition and growth factor withdrawal. In short, cells were plated in T75 or T175 flasks (according to experiment size), and growth medium was refreshed after 2 days. Once cells had reached full confluence, growth medium was once again replaced and contact inhibition was initiated for 24 h. Cells were then harvested, washed in HBSS twice, counted, and split at a 1:5 ratio into separate T25 or T75 flasks (according to experiment size) into mitogen-free medium (MCF10A medium without horse serum and EGF). After 24 h of growth inhibition, mitogen-free medium of all flasks except t = 0 was replaced with normal MCF10A growth medium. For each time point, cells were harvested from individual flasks. Double-thymidine block experiments were performed by plating cells in individual T25 flasks and adding thymidine at a concentration of 2.0 mM after 24 h (48 h for HCC1143). After 18 h of incubation, all flasks were washed with pre-warmed, CO_2_-equilibrated HBSS and allowed to progress in pre-warmed, CO_2_-equilibrated culture medium. After 8 h, thymidine was added again at a concentration of 2.0 mM and was removed by washing twice with pre-warmed, CO_2_-equilibrated HBSS after 18 h. Thymidine was not removed for the t = 0 time point, which was harvested directly. Of note, due to excessive cytotoxicity, MCF10A cells could not be reproducibly synchronized using double-thymidine. The double-thymidine block was also unsuitable for the synchronization of BT474 cells, which is likely due to their low proliferation rate [30]. Serum withdrawal experiments with MCF7 and MDA-MB-468 cells were performed by culturing cells for 24 h in normal growth medium, followed by washing twice with PBS, and further culturing in serum-deprived medium for up to 24 h. When applicable, medium was removed and replaced with normal growth medium for further sampling. For experiments involving palbociclib, MCF10A and BT474 were plated in separate flasks or plates for 24 h, followed by treatment with 1 µM palbociclib for up to 24 h. Palbociclib was then washed out by washing cells thrice with PBS, after which periodic sampling commenced. For all experiments, cells were harvested using Trypsin-EDTA (ThermoFisher), neutralized in an equal volume of Trypsin Neutralization Solution (Genlantis, San Diego, CA, USA), counted, and divided up in separate tubes according to the number of analyses. All centrifugation steps with live cells were performed at 450× *g* for 10 min, or 800× *g* for 15 min for final sample collection.

### 2.4. PI Staining, Flow Cytometry, and Fluorescence Microscopy

Single-cell suspensions were fixed in ice-cold 70% ethanol while vortexing and incubated on ice for at least 30 min. Cells were pelleted (450× *g*, 10 min), and ethanol was removed. Cells were washed twice in PBS before staining for 45 min with staining solution (PBS supplemented with 0.1% (*w*/*v*) Triton X-100, 20 µg/mL propidium iodide, and 200 µg/mL RNase A) at 1–2 × 10^6^ cells per m: at 37 °C. Cells were then analyzed immediately using the PE channel of a FACS Canto analyzer (BD Biosciences, Franklin Lakes, NJ, USA). To observe mitosis, live cells were stained with 0.3 µg/mL Hoechst in PBS for 5 min at room temperature, enclosed, and analyzed using conventional fluorescence microscopy. Fluorescent immunocytochemistry was performed using rabbit anti-53BP1, 1:1000 (ThermoFisher, PA1-16565).

### 2.5. Immunoblotting and mRNA Quantification

Immunoblotting and RT-qPCR were performed as described before [24]. The antibodies used for immunoblotting were rabbit anti-A3B, 1:1000 (5210-87-13 [26]); rabbit anti-Cyclin E2, 1:1000 (Cell Signaling, Danvers, MA, USA, #4132); rabbit anti-Cyclin B1, 1:1000 (Cell Signaling, D5C10); mouse anti-Cyclin A2, 1:2000 (Cell Signaling, BF683); rabbit anti-FoxM1, 1:1000 (Cell Signaling, D12D5); and mouse anti-Tubulin, 1:20,000 (Sigma-Aldrich, T5168). Primers used for RT-qPCR were as follows: *A3A*: FWD 5′-GAGAAGGGACAAGCACATGG-3′ and REV 5′-TGGATCCATCAAGTGTCTGG-3′; *A3B*: FWD 5′-GCCACAGAGAAGATTCTTAGCC-3′ and REV 5′-CGCCAGACCTACTTGTGCTA-3′; *CCNE2*: FWD 5′-TCCAAGAGTTTGCTTACGTCAC-3′ and REV 5′-GCCAGGAGATGATTGTTACAGG-3′; *CCNB1*: FWD 5′-TCTTGCAGTAAATGATGTGG-3′ and REV 5′-CAGTCAATTAGGATGGCTCT-3′; *B2M*: FWD 5′-CTTTGTCACAGCCCAAGATAG-3′ and REV 5′-CAATCCAAATGCGGCATCTTC-3′; *HPRT*: FWD 5′-TGACACTGGCAAAACAATGCA-3′ and REV 5′-GGTCCTTTTCACCAGCAAGCT-3′; *TBP*: FWD 5′-CCCATGACTCCCATGACC-3′ and REV 5′-TTTACAACCAAGATTCACTGTGG-3′.

## 3. Results

### 3.1. A3B Is Co-Expressed with Cell Cycle Proteins

To obtain a global overview of the A3B expression pattern in breast cancer cells, paraffin-embedded sections of an exploratory panel of breast cancer cell lines were subjected to immunohistochemistry (IHC) using a validated antibody against A3B [26,27,28]. While this antibody recognizes A3A, A3B, and A3G, detection of A3B is easily distinguished from A3A and A3G during microscopy and immunoblotting due to A3B’s distinct nuclear localization and size, respectively ([26,27,28]; discussed in [12]). We initially chose to investigate MDA-MB-415 and BT474, two cell lines known to express *A3B* at relatively high levels [5,31]. Two cell lines, *A3B*-null SKBR3 and *A3B*-low MDA-MB-330 [5,31,32,33], served as independent negative controls. While undetectable in these negative controls, A3B protein was detectable in the remaining cell lines and almost exclusively localized to the nuclear compartment, which is typical of this enzyme (Figure 1A) [4,34]. Intriguingly, the overall staining pattern in both A3B-high cell lines was far from uniform, with approximately 50% of cells almost completely lacking A3B signal.

Considering the major influence cell cycle progression exerts on gene expression genome-wide and the association of *A3B* with markers of proliferation in breast cancer (see Section 1), these initial observations prompted us to further consider the cell cycle as a major determinant of A3B expression. Thus, to differentiate between various phases of the cell cycle, a multiplex IHC assay was developed, which included the immunomicroscopic detection of A3B, Cyclin E2, Cyclin B1, and pan-Cytokeratin (pCK) (Appendix A, also see Section 2). Of note, for all analyses, subcellular localization was taken into account to ensure additional specificity; thus, nuclear staining of A3B and Cyclin E2 and cytoplasmatic staining of Cyclin B1 were quantified. Classically, Cyclin E2 expression peaks during the late G_1_/S phase and drops dramatically once G_2_ starts. Conversely, Cyclin B1 peaks during G_2_ to only enter the nuclear compartment during the relatively short mitotic phase (Figure 1B) [35,36]. Thus, these two cyclins are distinctly regulated and can be used as cell cycle markers. The multiplex assay was applied on asynchronous cultures (i.e., grown under normal tissue culture conditions) of various *A3B*-expressing breast cancer cell lines (BT474, HCC1599, HCC1954, and SUM225CWN; see [5,25]). As indicated above, Cyclin E2 is readily detectable in the nuclear compartment, and in separate cells, Cyclin B1 predominantly localizes to the cytoplasm. Furthermore, in agreement with our expectations, nuclear A3B is detectable in all expressing cell lines, but not in all individual cells (Figure 1C). Interestingly, individual breast cancer cells that lack A3B are often positive for Cyclin E2. 

Binomial testing, determining the probability of A3B being co-expressed with either Cyclin, was applied on single-cell images and further determined that A3B is significantly co-expressed (*p* < 0.001) with Cyclin B1, but not Cyclin E2 (*p* > 0.05; see Figure 1C). Additionally, nonparametric Spearman rank correlation revealed a statistically significant positive correlation between A3B and Cyclin B1 in all four cell lines (*p* = 0.664, 0.597, 0.669, 0.660, respectively; Figure 1D). Most cell lines also showed cells negative for both Cyclin E2 and Cyclin B1 (and A3B), which might represent quiescent cells [37]. Subsequently, specimens of treatment-naïve primary invasive ductal carcinomas were analyzed. In concordance with the observations in cell lines, most A3B-positive cells scored negative for Cyclin E2 and positive for cytoplasmic Cyclin B1 (Figure 1E,F). Combined, these results show that the expression of endogenous A3B in cancer cells is coordinated with proliferation and predominantly with the G_2_ and M phases of the cell cycle.

### 3.2. Endogenous A3B Expression Can Fluctuate throughout the Cell Cycle

Next, multiple cell synchronization methods were used to investigate the degree to which *A3B* expression changes throughout the cell cycle. Given the consistent observations amongst a diverse set of sample types thus far (i.e., multiple independent cell lines and tumor tissues), we opted to further diversify the cell line panel. Two additional breast cancer cell lines, MCF7 and HCC1143, as well as the previously featured HCC1954 cell line, were therefore included in cell cycle synchronization experiments. These cell lines represent the range of *A3B* expression found in clinical samples, with MCF7 expressing low levels, HCC1954 expressing moderate levels, and HCC1143 expressing high levels of *A3B* (Figure 2A; also see [5]). Additionally, these three cell lines were receptive to the same cell cycle synchronization method (Appendix A). Each cell line was synchronized with a double-thymidine block (see Section 2), and periodic samples were taken for RT-qPCR analysis upon release from the block. To complement the aforementioned multiplex IHC assay, gene expression of *CCNE2* (Cyclin E2) and *CCNB1* (Cyclin B1) was included to monitor cell cycle progression. 

Interestingly, after release from the double-thymidine block, *A3B* mRNA levels rise in all three cell lines, in a manner that associates positively with *CCNB1* expression and inversely with *CCNE2* expression (Figure 2B). However, it is notable that these trends are modest with the *A3B*-low cell line, MCF7A, intermediate in the *A3B*-medium cell line, HCC1954, and strongest with the *A3B*-high cell line HCC1143. In all instances, before the cultures reach peak *CCNB1* expression levels, a discernable increase in *A3B* expression is evident that is approximately 2-fold higher than expression levels at the G_1_/S border. 

Notably, for HCC1143, where protein levels are sufficiently high to be tracked by immunoblot, both A3B protein and mRNA levels closely associate and often peak slightly sooner in comparison to those of *CCNB1* throughout the entire cell cycle (Appendix A). 

In additional experiments, the normal-like mammary epithelial cell line MCF10A, which has very low baseline levels of endogenous *A3B* expression [38], was synchronized using contact inhibition and serum starvation (see Section 2). When released in a normal medium, MCF10A cells consistently lag in G_0_/early G_1_ for 6-9 h and reach the G_1_/S phase in 12 h, S phase in 18 h, and G_2_/M in 24 h, as judged by *CCNE2* and *CCNB1* expression (Appendix A). Other than an initial expression peak upon release in a normal medium (which likely is due to an effect of serum addition including many growth and signaling factors), the MCF10A cell line shows no appreciable changes in *A3B* during the bulk of its cell cycle (Figure 2C). Protein levels of A3B in these experiments tended to fluctuate, likely due to low signal-to-noise ratio and minor sampling differences between samples. These findings indicate that *A3B* expression in cancer cells may be coordinated with the cell cycle but is unable to do so in the normal-like cell line MCF10A, where expression of *A3B* is typically very low.

To obtain a view of *A3B* expression in the dormant G_0_ state, proliferation was halted in MCF7 cells by serum withdrawal, which is classically known to induce proliferative arrest [39,40]. Compared to the double-thymidine block, *A3B* expression is even more influenced by serum withdrawal, showing an ~80% decrease after 16 h (Figure 2D). Comparable observations were made in the cell line MDA-MB-468 which, after serum starvation, shows dramatically decreased levels of *A3B* as compared to published expression values [5]. These values steadily increase when MDA-MB-468 cells are released back into the normal medium (Appendix A). Thus far, the observations indicate that *A3B* expression in breast cancer cells requires proliferation and associates strongly with Cyclin B (*CCNB1* expression), indicative of the G_2_/M phase of the cell cycle. Additionally, and as defined previously [24], a repressive mechanism exists in normal-like MCF10A cells to prevent *A3B* from being expressed, even during active cell cycling and proliferation (discussed further below).

### 3.3. A3B Expression in Cancer Cell Lines Is Strongly Cell Cycle-Dependent

RNA sequencing expression profiles of *A3B* were associated previously with cell proliferation [22]. Additionally, earlier work showed that *A3B* transcription is repressed by the RB/E2F pathway [24,25], which itself is heavily influenced by cell cycle stage [41]. However, these earlier studies did not establish a mechanistic link between the cell cycle and *A3B* expression. To investigate whether the RB/E2F pathway controls A3B expression, MCF10A and BT474 cells were subjected to prolonged treatment with the CDK4/6 inhibitor palbociclib. Inhibition of the RB/E2F pathway through inhibiting CDK4/6 for 24 h has been shown to induce growth arrest in early G_1_ (Appendix A; also see [37]). The BT474 cell line was chosen for its considerable expression of *A3B*, which actively shapes the genome of this cell line in vitro [31,42,43], and MF10A cells were included as a normal-like control. Moreover, in our hands, palbociclib treatment is the only suitable synchronization method for these two cell lines (see Appendix A), allowing for the investigation of cell cycle dynamics in an RB/E2F context. In our experiments, each prolonged palbociclib treatment was followed by washout, release into normal medium, and periodic sampling for mRNA and protein analysis (Figure 3A). 

Synchronization in early G_1_ was successful in both cell lines, as measured by PI staining and flow cytometry (Figure 3B,C, upper panels). MCF10A cells swiftly resume proliferation after drug washout. Conversely, BT474 cells remain arrested for up to 18 h, during which near-complete early G_1_ arrest is obtained before cells begin to trickle into the S phase. Immunoblot analysis of cell cycle proteins, including Cyclin E2 and Cyclin B1, corresponded with the appropriate cell cycle phases (Figure 3B,C, lower panels). In MCF10A cells, A3B expression levels are relatively low, and protein levels drop below detection limits exclusively during growth arrest. However, following palbociclib washout, A3B expression levels increase and appear to track most closely with those of Cyclin B1 (Figure 3B, center panel). In comparison, *A3B* mRNA expression in BT474 is readily detectable and steadily declines during palbociclib treatment and subsequent early G_1_ arrest (Figure 3C, center panel). Importantly, although the decrease in A3B protein levels trails behind mRNA expression levels, both measures become significantly lower once a near-complete G_1_ arrest is obtained (Figure 3C). Given the observed prolonged growth arrest in BT474 even after palbociclib washout, an additional time course was set up that involved an extended outgrowth period. This experiment shows comparable results and additionally that A3B expression can revert to normal (i.e., high) levels at 72 h after palbociclib washout (Appendix A). Notably, in both cell lines, expression levels of A3B closely mirror those of the G_2_/M regulator FOXM1 which, like A3B, is repressed through the RB/E2F axis [24,44]. 

### 3.4. A3B Induction Dynamics throughout the Cell Cycle

Several mechanisms that activate *A3B* transcription depend on the activation of the inflammatory PKC/ncNF-κB pathway [38,45,46,47,48,49]. This pathway, in turn, is activated by various intra- and extracellular stimuli, such as DNA damage from ionizing radiation (IR), chemical interference with pathway components, and/or ligand-based receptor activation (reviewed in [12,50]; also see [38]). It is therefore mechanistically insightful to know in which cell cycle phase the PKC/ncNF-κB pathway is most capable of inducing *A3B*. For instance, because A3B preferentially targets ssDNA [51,52], which is more abundant in actively dividing cells as compared to growth-arrested cells, induction during proliferation may be particularly mutagenic.

In order to efficiently investigate the cell cycle dynamics of *A3B* expression, we first set out to select cell lines with differential responses to induction. For this purpose, the chemical compound phorbol myristic acid (PMA) was used to induce PKC/ncNF-κB-dependent expression of *A3B* in MCF10A and HCC1143. MCF10A cells show high *A3B* inducibility, whereas HCC1143 breast cancer cells, which have much higher baseline levels of endogenous *A3B*, show appreciably lower *A3B* induction potential (Figure 4A, Appendix A). We propose that the PKC/ncNF-κB pathway and the *A3B* promoter may already be partially activated in HCC1143 cells. Indeed, earlier work has shown that the PKC/ncNF-κB pathway in many cancer cell lines is constitutively activated during culture and is actively driving *A3B* expression [38]. MCF10A was synchronized and cells in various cell cycle stages were treated with PMA (Figure 4B). Cell cycle progression was analyzed by flow cytometry and RT-qPCR, and kinetics comparable to earlier experiments were obtained (compare Figure 4C to Appendix A). As a control, the expression of Cyclin E2 was analyzed since PKC/ncNF-κB activation is known to repress *CCNE2* [53] (Appendix A). Importantly, A3B was induced minimally in G_0_-arrested cells, indicating that potent repression is present in this cell cycle phase even when PKC/ncNF-κB pathway activation is strong (Figure 4D). 

Interestingly, A3B was induced to higher levels in cells released from G_0_ arrest (Figure 4D). An independent time course yielded comparable results (Appendix A). To control for the possibility that the relative lack of A3B inducibility during G_0_ is due to a starvation-induced decrease in viability and signal responsiveness, A3A expression was analyzed in parallel. Like A3B, A3A has been found to be inducible by PMA and therefore served as a potent control for induction [45]. Importantly, the immediate induction of A3A by PMA in G_0_-arrested cells indicates that the relative lack of A3B induction throughout G_0_ is not an artifact of our synchronization method (Appendix A).

We also analyzed whether the strong induction of A3B during proliferation could lead to DNA damage, especially considering the association of A3B with DNA strand breaks and the activation of the DNA damage response [5]. We therefore treated growth-arrested and actively cycling cells with PMA and analyzed the nuclear accumulation of the DNA damage marker 53BP1 by immunofluorescence microscopy. Contrary to our expectations, treatment with PMA did not lead to the induction of 53BP1 foci under any condition tested (Appendix A). An explanation for this observation is discussed below.

So far, the results show that *A3B* transcriptional activation through the PKC/ncNF-κB pathway is influenced by cell cycle progression and that induction is impaired in G_0_. Repression of cell cycle-associated genes is often facilitated by E2F-based transcriptional complexes, including the DREAM complex, which predominantly represses genes throughout the dormant G_0_ phase (reviewed in [41]). We have previously shown that E2F-based complexes also repress *A3B* transcription in normal mammary epithelial cells by binding to an *A3B* resident E2F site [24]. To investigate whether this site facilitates repression of *A3B* specifically during G_0_, MCF10A cells engineered to lack the E2F site through CRISPR/Cas9-mediated base substitution (HDR clones, published in [24]) were synchronized and treated with PMA. The cell line MCF10A-HDR53 was used; similar to other HDR clones, it expresses increased levels of A3B as compared to controls (Appendix A). Both control and HDR cell lines arrested effectively at G_0_ and reached the G_1_/S phase in 15 h, as indicated by a sharp increase in *CCNE2* (Figure 4E). Intriguingly, while the A3B induction pattern in the control cell line corresponds to earlier results (i.e., low induction in G_0_ and high induction after, compare Figure 4D,F), induction of A3B in G_1_/S-phase cells is only marginally higher than that in G_0_ in the MCF10A-HDR53 cell line. Thus, the transcriptional activation of *A3B* by the inflammatory PKC/ncNF-κB pathway is most efficient when cells are actively proliferating and is actively repressed by E2F repressive complexes when cells are arrested in G_0_ and/or early G_1_. Additionally, a comparison of data from the 0 and 15 h timepoints suggests that A3B expression levels may remain stable throughout the cell cycle in cells lacking this single E2F site (Figure 4F, light blue bars of HDR53). Combined with the findings mentioned above, these data show that *A3B* is universally repressed by the RB/E2F pathway throughout G_0_ in normal and cancer cells alike. The gain-of-function experiments in our engineered MCF10A cells also suggest that there may be no other repressive mechanisms during all other cell cycle phases and that aberrant pathway activation during proliferation likely explains A3B overexpression observed in a significant proportion of breast cancers.

## 4. Discussion

This study constitutes the first detailed insights into the cell cycle dynamics of A3B expression in breast cancer. Starting with the observation that A3B expression is heterogeneous in cultured breast cancer cells, we show that A3B expression associates strongly with Cyclin B1 and not Cyclin E2 on a single-cell level in cell lines and clinical breast cancer specimens. Furthermore, breast cancer cell line synchronization studies indicate that *A3B* expression levels oscillate throughout the cell cycle in a manner that associates positively with Cyclin B1 (G_2_/M phase) and negatively with Cyclin E (G_0_/G_1_ phase). Surprisingly, expression and induction of A3B during growth arrest are tightly repressed by the RB/E2F pathway in all cell lines tested. Conversely, A3B induction is a prominent feature during cell cycle progression of A3B-overexpressing breast cancer cells, and only possible during proliferation in cells with low constitutive expression. Thus, our study offers new insights into the mechanism of A3B regulation during the cell cycle. Building on our previous work [24,25], we propose that a general lack of RB/E2F-mediated repression does not fully explain the observed dysregulation of A3B expression in breast cancers. Instead, the combined effects of RB/E2F de-repression and PKC/ncNF-κB pathway activation appear to be required to achieve the high A3B expression levels observed in many cancer cell lines and tumors (model in Figure 5). This model is also supported by work from other groups, which indicated that A3B expression is associated with proliferation [21,22,23,54].

Over the recent years, various publications have reported on the transcriptional regulation of *A3B*. Since its first association, the PKC/ncNF-κB pathway has continued to take a central role in *A3B* regulation [38,47,55]. The PKC/ncNF-κB pathway conveys intracellular and extracellular signals such as DNA damage and TNFα, respectively, and thus relays cellular stress and inflammatory signals to the *A3B* promoter [38,47,49]. As seen in this study and others, the PKC/ncNF-κB pathway and, by extension, the expression of *A3B* can also be stimulated directly using PMA. Conversely, transcriptional *repression* is facilitated by the RB/E2F pathway and is enforced by the binding of E2F-based repressive complexes to an E2F site within the *A3B* promoter [24,25]. Disruption of this pathway raises A3B expression in a way that mirrors the action of viral oncogenes E7 and polyomavirus large T antigen, which are well known for their ability to induce *A3B* during viral infections [25,29]. The data presented in this work dovetail with this prior knowledge and indicate that when these regulatory mechanisms are no longer in homeostasis, a “perfect storm” scenario might unfold. In this scenario, the combination of DNA damage, inflammation, and a generally pro-proliferative cellular state raise A3B protein levels to a degree where they can actively influence disease trajectory.

Three findings, in particular, were unexpected in this study. First, whereas in the current study a 24 h exposure to palbociclib yielded a potent downregulation of A3B, earlier work showed that inhibition of CDK4/6 using palbociclib does not directly influence A3B expression [25]. At the time, it was proposed that a mechanism secondary to the RB/E2F pathway might also regulate *A3B* expression. We now show that a decrease in *A3B* expression coincides with arrest in G_0_/early G_1_, strongly implicating proliferative status with the expression dynamics of *A3B*. In follow-up research, it would be informative to include a transcription factor screen of the *A3B* promoter during palbociclib treatment and during promoter activation in response to PMA, which would further elucidate the transcriptional dynamics of *A3B*. Secondly, repression of *A3B* during G_0_ appears to be largely intact in all cell lines included in this study, including cells with high expression of *A3B*. We previously proposed that overexpression of A3B, in part, may be attributable to the failure of some breast cancer cells to repress *A3B* through the RB/E2F pathway. Although this is still supported by multiple lines of evidence, the current study offers more detailed insights and indicates that cancer cells with high A3B expression reside disproportionally in cell cycle phases where endogenous and exogenous stimuli are free to activate the promoter. Thus, we propose that overexpression of A3B is the culmination of untimed proliferation and activation of (inflammatory) pathways, including the PKC/ncNF-κB pathway (Figure 5). Finally, it was expected that increased expression of A3B in response to PMA would mount a DNA damage response. The absence of this response may be explained by the possibility that PMA interferes directly with the cell’s ability to mount a DNA damage response. As also seen in our studies, acute administration of PMA severely slows cell cycle progression [39,53]. Thus, by reducing ssDNA present at replication forks, cytosine deamination and the associated DNA damage response are avoided [45]. Alternatively, the levels of A3B reached with PMA may not be high enough to inflict serious DNA damage in our model. In either scenario, we propose the use of other techniques, such as CRISPR activation or lentiviral transduction at high MOI, to investigate the effects of cell cycle-specific APOBEC activity on the DNA damage response. Combining these efforts will provide valuable insights into how A3B-stimulating pathways and the cell cycle of tumor cells combine to induce A3B, providing the mutagenic fuel that drives tumor evolution.

The clinical relevance of this work is at least two-fold. First, it explains the heterogeneity observed in tumor A3B protein expression by IHC, with strong positivity associating with active cell cycling and specifically with the G_2_/M phase of the cell cycle (coincident with Cyclin B1 positivity). Such tumors are not only actively dividing but are also therefore likely to be undergoing active DNA deamination, damage, and mutagenesis caused by A3B (i.e., ongoing evolution). This finding is corroborated by other work that shows that A3B is preferentially expressed during G_2_/M in myeloid cells [23]. Since APOBEC can synergize with DNA damage-inducing drugs across multiple cancer types, including breast and ovarian cancer [56,57], it may be beneficial to combine such treatments with G_2_-arresting drugs. Inducing G_2_ arrest in A3B-positive tumors to be treated with IR may also provide clinical benefit since tumor cells are most sensitive to DNA damage in G_2_, during which A3B expression may be induced further to possibly toxic levels. Future cell-based assays should determine the dosage, timing, and feasibility of such an approach. Second, this work shows the applicability of multiplex IHC with an A3B antibody and two cell cycle proteins. In essence, any other clinically relevant marker may be co-stained using this approach, which may then inform the tumor microenvironment of A3B-expressing tumors. For example, the infiltration of CD4^+^/CD8^+^ T-cells into A3B-expressing tumor mass could be directly observed in clinical biopsies. This is exceptionally relevant since recent evidence indicates that APOBEC-dominated tumors might be receptive to immune-based anticancer therapies [58,59,60].

## 5. Conclusions

In conclusion, the data presented here consolidate the proposed association between cellular proliferation and A3B expression and offer further insights into the cell cycle dynamics of the regulation of this cancer genomic DNA deaminase. In breast cancers, A3B overexpression appears to be the cumulative result of proliferation-associated relief from repression with concomitant pathway activation (Figure 5).

## Figures and Tables

**Figure 1 cells-12-01185-f001:**
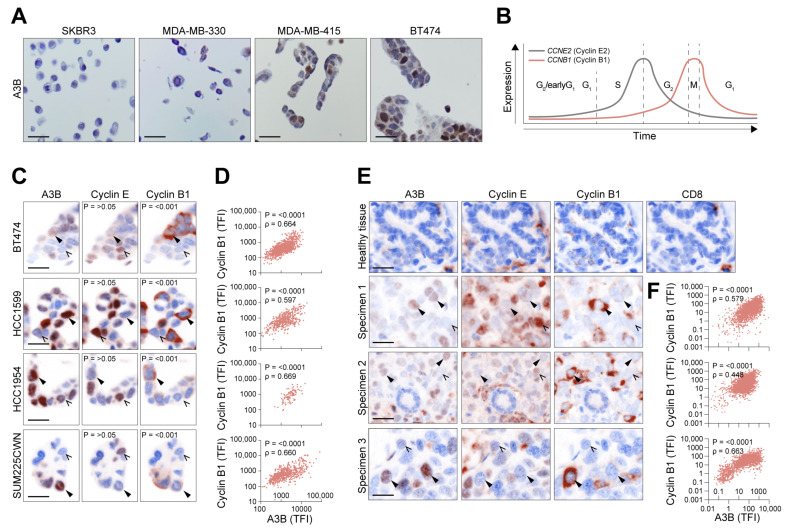
Protein expression of A3B in breast cancer is heterogeneous and correlates with cell cycle proteins. (**A**) Photomicrographs of A3B using conventional IHC on a diverse set of breast cancer cell lines, including the A3B-null (SKBR3), A3B-low (MDA-MB-330), and A3B-high (MDA-MB-415 and BT474) cell lines. Note the heterogeneous expression pattern of A3B in MDA-MB-415 and BT474. (**B**) General overview of the various cell cycle phases, overlaid with proteins known to peak during each respective phase. G_0_/G_1_ is marked by low expression of both cyclins, while *CCNE2* (Cyclin E2) expression rises throughout G_1_ and into G_1_/S. Expression of CCNB1 (Cyclin B1) increases during G_2_ and peaks at mitosis. (**C**) Photomicrographs showing the subcellular localization of A3B, Cyclin E2, and Cyclin B1 in various A3B-high breast cancer cell lines. Images acquired by multiplex IHC staining. Solid arrowheads indicate cells expressing A3B; open arrows point out cells negative for A3B. P-values are calculated by binomial testing, determining the probability of A3B being co-expressed with individual cell cycle markers. (**D**) Quantification of (subcellular specific) total fluorescent intensity (TFI) of individual cells as in (**C**), using Spearman correlation. (**E**,**F**) Healthy tissue and three primary breast cancer tumors stained for A3B, Cyclin E2, and Cyclin B1, using the same workflow and analysis presented in (**C**,**D**). CD8 staining of healthy tissue is included to account for the sporadic cytoplasmic staining encountered in isolated cells, attributable to A3G expressing tumor-resident leukocytes. Note that detection of A3B protein through regular and multiplex IHC requires separate optimization and antibody concentrations, accounting for the slight difference in staining pattern between these two procedures. Scale bars denote 20 µm. TFI: total fluorescent intensity.

**Figure 2 cells-12-01185-f002:**
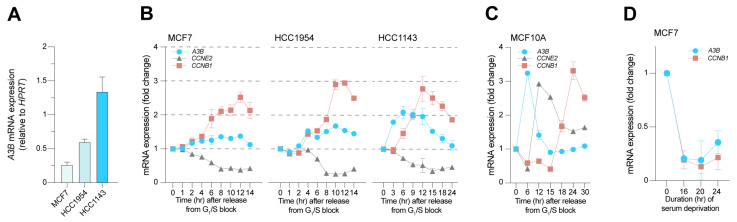
*A3B* expression in relation to cell cycle markers *CCNE2* and *CCNB1*. (**A**) Levels of endogenous *A3B* mRNA expression during normal culture of A3B-low (MCF7), A3B-intermediate (HCC1954), and A3B-high (HCC1143) breast cancer cells (n = 3). (**B**) Expression of *A3B*, *CCNE2*, and *CCNB1* in MCF7, HCC1954, and HCC1143 released from G_1_/S arrest, induced by a double-thymidine block. The peak in *CCNB1* expression denotes the G_2_/M phase, whereas *CCNE2* drops soon after release since *CCNE2* expression is high throughout G_1_. The different sampling times are chosen to accommodate the differences in the growth speed of the used cell lines. Error bar represents SD. MCF7 and HCC1954 data represent n = 2 independent experiments; HCC1143 data represent n = 6 independent experiments. (**C**) Cell cycle progression and *A3B* expression in the normal-like breast epithelial cell line MCF10A, which expresses very low levels of *A3B*. Data represent n = 2 independent experiments. (**D**) Gene expression levels of *A3B* and *CCNB1* as measured in MCF7 breast cancer cells after varying lengths of serum deprivation. Values relative to normal growth conditions. Error bar represents SD. Data represent n = 2 independent experiments.

**Figure 3 cells-12-01185-f003:**
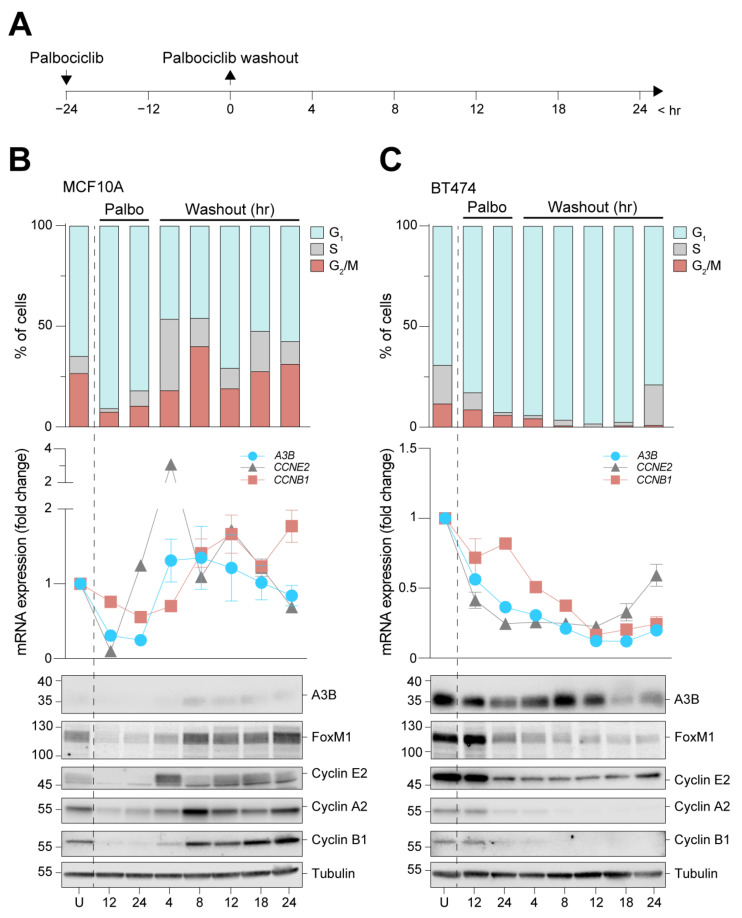
A3B is repressed during G_0_/early G_1_ arrest in normal-like MCF10A cells and A3B-high BT474 cells. (**A**) Palbociclib arrests MCF10A and BT474 cells in G_0_/early G_1_ by direct interference with the RB/E2F pathway. Cells were treated for 24 h to induce growth arrest, followed by washout and subsequent sampling. (**B**,**C**) Cell cycle progression (top panels) and RT-qPCR and immunoblot analysis (center and bottom panels, respectively) of A3B, Cyclin E2, and Cyclin B1 during and after palbociclib-induced growth arrest of MCF10A and B474 cells. For immunoblots, samples of both cell lines were loaded on the same gel, but data are depicted separately here due to image intensity adjustments required for visualization of A3B in each cell line. U = unsynchronized. Error bar represents SD. Data represent n = 4 independent experiments.

**Figure 4 cells-12-01185-f004:**
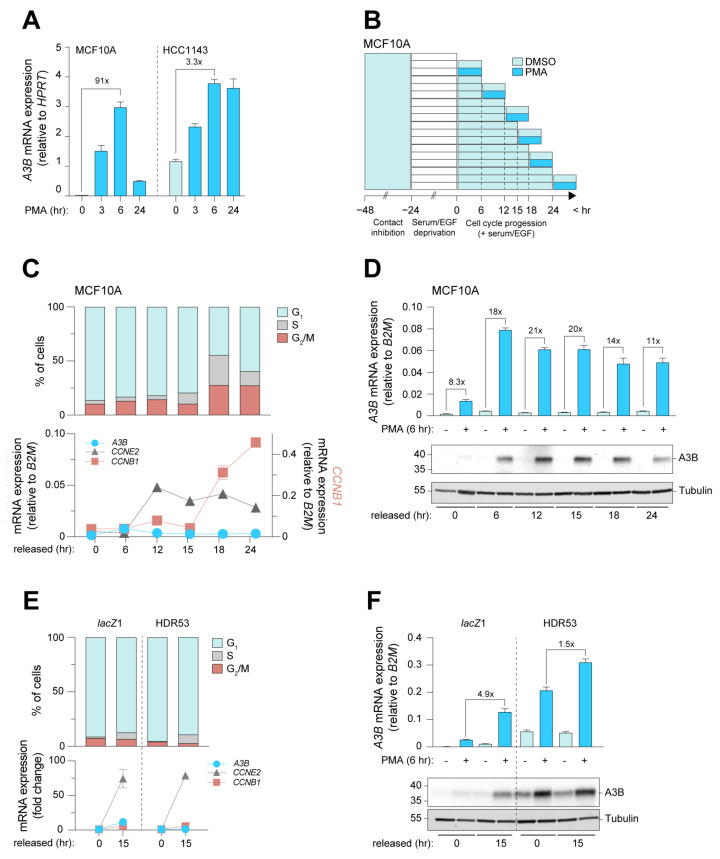
Induction of A3B expression is strongest in actively proliferating cells, and repression during G_0_ is RB/E2F-dependent. (**A**) Analysis used to determine the most suitable model for the investigation of PKC/ncNF-κB-induced expression of *A3B* within the context of cell cycle progression. Unsynchronized MCF10A and HCC1143 cells were exposed to PMA for up to 24 h, during which samples were taken for RT-qPCR analysis. Error bar represents SD. Data represent n = 2 independent experiments. (**B**) Method used to induce A3B expression using PMA at defined cell cycle stages of MCF10A cells. Teal denotes contact inhibition, followed by growth factor withdrawal (white), cell cycle progression (light blue), and treatment with PMA or vehicle. Samples taken at the start of each PMA or vehicle treatment served as independent controls for cell cycle progression. (**C**) Cell cycle progression of MCF10A as analyzed by PI staining followed by flow cytometry, and RT-qPCR analysis of *A3B*, *CCNE2,* and *CCNB1*. Error bar represents SD. Data represent n = 2 independent experiments. (**D**) A3B mRNA (top) and protein (bottom) expression upon PMA treatment during indicated time points following synchronization. Error bar represents SD. Data represent n = 2 independent experiments. (**E**) Synchronization of *lacZ* control and MCF10A-HDR53 cells, as analyzed by PI stain and RT-qPCR of *A3B*, *CCNE2*, and *CCNB1*. Gene expression is given as fold change over baseline relative to the cell cycle housekeeping gene *B2M*. Error bar represents SD. Data represent n = 3 independent experiments. (**F**) Induction of A3B upon PMA treatment in G_0_-arrested and actively cycling *lacZ* control and MCF10A-HDR53 cell lines. Fold changes above the brackets indicate the post-induction levels of *A3B* in arrested cells with those in actively cycling cells. Error bar represents SD. Data represent n = 3 independent experiments.

**Figure 5 cells-12-01185-f005:**
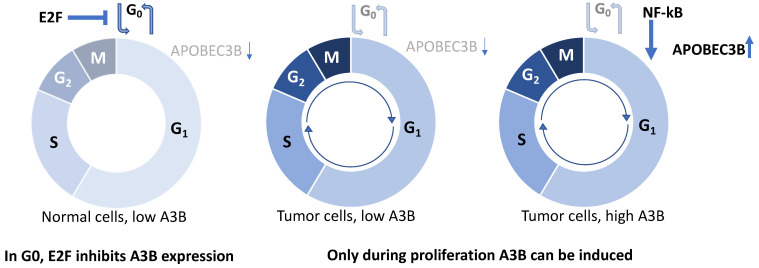
Model explaining current results. In normal, non-proliferating cells in G_0_, E2F inhibits *A3B* gene expression (left). During proliferation, such as in rapidly dividing tumor cells, E2F no longer inhibits *A3B* expression (middle), yet for *A3B* to be overexpressed in a subset of tumors, stimulation during proliferation is required through the NF-kB pathway (right).

## Data Availability

Data and materials are available upon request to P.N.S.

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
