# Peer review of "Aberrant APOBEC3B Expression in Breast Cancer Is Linked to Proliferation and Cell Cycle Phase"

_cells, 2023, doi:10.3390/cells12081185_

Round 1

Reviewer 1 Report

The study investigates the association between A3B expression and cell cycle progression in breast cancer. The authors find that A3B expression fluctuates throughout the cell cycle in cancer cells.  It is credible for further understanding the

APOBEC3B function. However, there are many limitations as followed.

 1.       In the Introduction part, the importance of the APOBEC family in breast cancer was not well presented.

2.       The study did not explain the clinical relevance of A3B overexpression in cancer patients. 

3.      The study mainly explore the relationship between A3B expression and different cyclins levels. However, the mechanism is not well presented and needed to be further investigated.

4.       There are no references for many previous results, such as in line 202-203 ‘While this antibody recognizes A3A, A3B, and A3G, detection of A3B is easily distinguished from A3A and A3G during microscopy and immunoblotting, due to A3B’s distinct nuclear localization and size.’

5.      The number of cell lines in this paper is very complicated. The authors should add a figure about A3B expression in a different cell lines.(all breast cells used in this paper).

6.      The authors used many cell lines in their work without any reason. It is confused.

7.      In Figure 3B, the tublin expression of WB image was inconsistent.

8.      In figure 1C, the IHC on A3B in BT474 differed significantly from the result in Figure1A.

9.      The authors should discuss more about expression pattern in different cells. Some of the cells lines with high staining, while with approximately 50% of cells almost completely lacking A3B signal.”

10.   Figure4A and D, the relative RNA level is not well presented. The control level is not constant.

11.   Figure5, all the periods of the G1, S, G2 and M are the same, which is not suitable.

Author Response

The study investigates the association between A3B expression and cell cycle progression in breast cancer. The authors find that A3B expression fluctuates throughout the cell cycle in cancer cells.  It is credible for further understanding the APOBEC3B function. However, there are many limitations as followed.

  1. In the Introduction part, the importance of the APOBEC family in breast cancer was not well presented.
  2. The study did not explain the clinical relevance of A3B overexpression in cancer patients.

Response: We agree that the importance of the APOBEC family in breast cancer may have been underexposed in the introductory paragraphs. We have therefore added more details regarding this topic (see marked changes in the introduction of the revised version), which also includes explicit mention of several cancer driving mutations that are attributable to APOBEC activity. Additionally, we recently published a review on this topic (Roelofs et al., Clin Cancer Res 2022, new ref 12), which has now been added as a reference and addresses this issue in greater detail.

Furthermore, we realized that the text referring to the clinical relevance of A3B also included implicit mention of A3A, which may have been a confusing approach, if this is what the reviewer means with his/her comment. We have now streamlined this text in the introduction to specifically address the relevance of A3B in breast cancer patients.

  1. The study mainly explores the relationship between A3B expression and different cyclins levels. However, the mechanism is not well presented and needed to be further investigated.

Response: The mechanism by which APOBECs are induced or repressed has been studied in detail in numerous papers referenced in the current manuscript. We included for example PMID: 26420215 (Leonard et al. Cancer Res 2015) on the induction by the NF-kB pathway, and have recently shown how in G0 A3B expression is repressed via E2F in PMID: 32985974 (Roelofs et al. eLIFE 2020). Our current study clarifies how these mechanisms pertain to the observation of heterogenous expression of APOBEC3B in breast cancer cell lines and tissues, and highlight how mis-regulation of these mechanisms may result in aberrant A3B overexpression in breast cancer, which we clarify in figure 5. Furthermore, we have made substantial changes throughout the manuscript and abstract in order to be more clear on the merit and message of our data. Most importantly, in response to this comment we have decided to change the title to “Aberrant APOBEC3B expression in breast cancer is linked to proliferation and cell cycle phase” which believe better reflects our findings.

  1. There are no references for many previous results, such as in line 202-203 ‘While this antibody recognizes A3A, A3B, and A3G, detection of A3B is easily distinguished from A3A and A3G during microscopy and immunoblotting, due to A3B’s distinct nuclear localization and size.’

Response: We appreciate the reviewer’s comment and have re-read the document to find statements that lack proper referencing. The statement regarding the 5210-87-13 antibody was already supported by the reference to Brown et al. in the previous sentence, to which this sentence connects contextually. To provide more clarity we have now added a reference to our recent review (Roelofs et al., Clin Cancer Res. 2022) which includes a specific reference to the application of this antibody. Throughout the manuscript we have added references to other statements.

  1. The number of cell lines in this paper is very complicated. The authors should add a figure about A3B expression in a different cell lines. (all breast cells used in this paper).
  2. The authors used many cell lines in their work without any reason. It is confused.

Response: We acknowledge that a large number of cell lines was used here because each has helped to test a particular aspect of our central hypothesis as well as to generalize some of the major findings. Firstly, in an “exploratory panel” of cell lines (and tumors) variability of A3B expression and correlation with cyclins was investigated. Next, several cell lines were used that exhibited different baseline A3B expression levels: in some cell lines, no A3B expression was found, even during proliferation. However, induction of A3B expression using PMA is possible in these cell lines, although notably not during G0. Other cell lines show intermediate or abundant A3B expression levels, and by using different cell cycle synchronization methods, we find that these only express A3B during proliferation. Limited additional induction -and only outside G0- was possible in the highly expressing cell lines. We believe that the use of multiple cell lines is required to ensure robust testing of our central hypothesis (as well as appropriate extension and validation in human breast cancer tissues), and that together these different cell line and tissue experiments clarify the underlying mechanism.

We do, however, acknowledge this may be confusing. Therefore, to improve the readability of our manuscript, we have added a supplemental table that outlines the main cell lines used (MCF10A, MCF7, HCC1954, BT474, and HCC1143) and their relevant characteristics, including A3B expression, available cell cycle synchronization methods, and their A3B inducibility by PMA. We hope that this table provides a clearer overview of the reasons why the study includes a diverse panel of cell lines. We have also cited this table throughout the manuscript and explained in several places why these cell lines were used.

Additionally, we now refer to the SKBR3, MDAMB330, MDAMB415, and BT474 as an “exploratory panel” of cell lines. In fact, this staining represents one of our earliest findings that A3B expression may be potently repressed during at least one cell cycle phase in breast cancer cells, and therefore very much represents an exploratory analysis.

  1. In Figure 3B, the tubulin expression of WB image was inconsistent.

Response: Thank you for pointing this out. The tubulin expression of Figure 3B is consistent, however, we agree that the tubulin expression of Figure 3C was not consistent and showed unexpected doublet bands. The Cyclin E2 blot used here showed the same issue. We have added blots of a repeated run using the same samples, which show greater consistency.

  1. In figure 1C, the IHC on A3B in BT474 differed significantly from the result in Figure1A.

Response: Thank you very much for pointing this out, as we recognize that this may need elaboration. The IHC images in Figure 1A were obtained through conventional IHC, whereas the images in Figure 1C were obtained by multiplex IHC (Vectra). Since these two procedures require separate optimization procedures for A3B, the staining patterns differ to a certain extent. Given the distinct experimental background, these two methods should be viewed as separate entities and cannot be directly compared. We have now added text that explicitly separates these two procedures.

  1. The authors should discuss more about expression pattern in different cells. Some of the cells lines with high staining, while with approximately 50% of cells almost completely lacking A3B signal.”

Response: Please see also response #5 above. While it is correct that cell lines with overall high A3B expression can still show large proportions of cells that lack expression, we believe that this, in fact, is one of the main early observations made (and addressed) in this study. We observed that the A3B expression in cancer cells can still be low if these individual cells are arrested in G0, while in proliferative cell cycle phases A3B can be induced through non-canonical (nc)NF-kB signaling pathway (and de-repression of the RB-E2F pathway). The ncNF-kB pathway fails to activate A3B when cells are arrested in G0, see figure 5. Therefore, the expression of A3B appears as a function of timed transcriptional activation throughout later cell cycle phases, rather than a function of the cancerous nature of cell lines. We have made changes throughout the manuscript to convey this matter more clearly, including the title change.

  1. Figure 4A and D, the relative RNA level is not well presented. The control level is not constant.

Response: We appreciate this feedback. We want to point out that HPRT was used as a housekeeping gene in Figure 4A since this gene is routinely used as a reference gene (https://pubmed.ncbi.nlm.nih.gov/15543203/), and indeed was relatively stable after PMA treatment  in vitro. However, HPRT is not suitable as a housekeeping gene in MCF10A cell cycle synchronization experiments, for which B2M proved more suitable in our hands. Since “relative expression” as opposed to “fold change” in panel 4D provided the clearest overview of our results we opted to use this here. Since Figure 4A and 4D serve to different purposes (expression differences between cell lines and A3B inducibility through the cell cycle, respectively), we opted to use and name both housekeeping genes. We have also updated some of the Y-axes of our RT-qPCR analyses to more adequately separate them from immunoblot analyses.

  1. Figure 5, all the periods of the G1, S, G2 and M are the same, which is not suitable.

Response: We have adjusted the time spans of the different cell cycle phases to better reflect reality.

Reviewer 2 Report

This study investigates the cell cycle-dependent regulation of APOBEC3B (A3B) expression in breast cancer cells using several cell lines and suggests that the pathways of RB/E2F and PKC/non-canonical NF-kB are involved. Although the conclusion is true in some cell settings, whether it can be a general regulatory mechanism in most cells is uncertain. It may need more evidence.

It is better to illustrate the regulatory mechanism of A3B expression using consistent cancer cell lines, because the characteristics of cancer cells are easily changed and evolved during cell culture. This study used different cancer cell lines in Figs 1, 2, 3 and supplementary Figures, which may not faithfully reflect a complete story for each individual cell line. As the Ben-David’s paper (PMID: 30089904) shows that the 27 MCF7 cell strains obtained from different labs exhibit diverse phenotypes and show differential or opposite responses to the same drug treatments, the expression of A3B may also be regulated in a cell-dependent manner.

Again, the sensitivity to Palbociclib or to other drugs is cell-dependent, which is more obviously different between cancer and non-cancer cells (Fig 3). Therefore, high-dose of Palbociclib may cause adverse effect to the cancer cells and influence the recovery from cell cycle arrest or the regulation of A3B expression. It may be better to treat the cells by the doses depending on their sensitivity to Palbociclib. Fig 3C, the quantitative data (middle panel) and blot image of A3B are not matched, the blot shows increased A3B at 8h, 12h but the quantitative data are decreased.

In Fig S2B, the A3B protein was induced in concordance with the induction of cyclin E mRNA and was further increased upon the appearance of cyclin B mRNA in MCF10A cells (n=?). This pattern is similar as that in cancer cells and does not completely fit to the statement in lines 308-309 (different A3B expression patterns in cancer and non-cancer cells). Another study (PMID: 33592502) also showed that A3B mRNA were enriched in G2/M-phase in normal blood cells.

Considering the coordination of RNA and protein expression levels, whether A3B protein and/or mRNA modifications affect their amounts across various cell cycle phases and different cell lines?

Author Response

Reviewer 2

  1. This study investigates the cell cycle-dependent regulation of APOBEC3B (A3B) expression in breast cancer cells using several cell lines and suggests that the pathways of RB/E2F and PKC/non-canonical NF-kB are involved. Although the conclusion is true in some cell settings, whether it can be a general regulatory mechanism in most cells is uncertain. It may need more evidence. It is better to illustrate the regulatory mechanism of A3B expression using consistent cancer cell lines, because the characteristics of cancer cells are easily changed and evolved during cell culture. This study used different cancer cell lines in Figs 1, 2, 3 and supplementary Figures, which may not faithfully reflect a complete story for each individual cell line. As the Ben-David’s paper (PMID: 30089904) shows that the 27 MCF7 cell strains obtained from different labs exhibit diverse phenotypes and show differential or opposite responses to the same drug treatments, the expression of A3B may also be regulated in a cell-dependent manner.

Response: We appreciate this point and acknowledge that a large number of cell lines was used here because each has helped to test a particular aspect of our central hypothesis as well as to generalize some of the major findings. Firstly, in an “exploratory panel” of cell lines (and tumors) variability of A3B expression and correlation with cyclins was investigated. Next, several cell lines were used that exhibited different baseline A3B expression levels: in some cell lines, no A3B expression was found, even during proliferation. However, induction of A3B expression using PMA is possible in these cell lines, although notably not during G0. Other cell lines show intermediate or abundant A3B expression levels, and by using different cell cycle synchronization methods, we find that these only express A3B during proliferation. Limited additional induction -and only outside G0- was possible in the highly expressing cell lines. We believe that the use of multiple cell lines is required to ensure robust testing of our central hypothesis (as well as appropriate extension and validation in human breast cancer tissues), and that together these different cell line and tissue experiments clarify the underlying mechanism.

We do, however, acknowledge this may be confusing. Therefore, to improve the readability of our manuscript, we have added a supplemental table that outlines the main cell lines used (MCF10A, MCF7, HCC1954, BT474, and HCC1143) and their relevant characteristics, including A3B expression, available cell cycle synchronization methods, and their A3B inducibility by PMA. We hope that this table provides a clearer overview of the reasons why the study includes a diverse panel of cell lines. We have also cited this table throughout the manuscript and explained in several places why these cell lines were used.

Additionally, we now refer to the SKBR3, MDAMB330, MDAMB415, and BT474 as an “exploratory panel” of cell lines. In fact, this staining represents one of our earliest findings that A3B expression may be potently repressed during at least one cell cycle phase in breast cancer cells, and therefore very much represents an exploratory analysis.

  1. Again, the sensitivity to Palbociclib or to other drugs is cell-dependent, which is more obviously different between cancer and non-cancer cells (Fig 3). Therefore, high-dose of Palbociclib may cause adverse effect to the cancer cells and influence the recovery from cell cycle arrest or the regulation of A3B expression. It may be better to treat the cells by the doses depending on their sensitivity to Palbociclib.

Response: Thank you for expressing your concern regarding cell type-dependent responses to Palbociclib. We do agree that cell lines can typically portray vastly different responses to certain drugs. However, we reported in Starrett et al. (mBio, 2019) that, with the exception of cell lines with known defects in the RB/E2F pathway (such as HCC1937), cell lines generally respond well to a short Palbociclib treatment. Even those cell lines with RB/E2F defects showed a minor response upon 4 hours of exposure, which prompted us to extend the Palbociclib treatment to 24 hours during our current study. We also chose to include more controls in our current assay, including several other Cyclins and a PI-stain to ensure accurate observation of cell cycle arrest and, at least, the first period of cell cycle re-entry.

We do agree that BT474 cells recover from cell cycle arrest differently (i.e., slower) than MCF10A cells. However, this is much more likely due to the low proliferation rate of BT474 than its responsiveness to Palbociclib, which is emphasized in Supplemental Figure. 3B. Thus, although differences between MCF10A and BT474 are evident, we are confident that a sufficient degree of CDK4/6 inhibition was reached to inform on the RB/E2F-mediated repression of A3B expression throughout G0/early G1 in both cell types.

  1. Fig 3C, the quantitative data (middle panel) and blot image of A3B are not matched, the blot shows increased A3B at 8h, 12h but the quantitative data are decreased.

Response: We appreciate your feedback. The middle panel of Figure 3C displays RT-qPCR data and thus reports on gene expression, while the lower panel reports on protein expression. The dynamics of RNA and protein are -logically- different. We have realized that the description of this figure lacked some crucial elements regarding this distinction. In our current version of the manuscript, we have updated the Y-axis and legend to more accurately describe the different experimental approaches. We have also changed the Y-axes of other figures in order to maintain consistency.

  1. In Fig S2B, the A3B protein was induced in concordance with the induction of cyclin E mRNA and was further increased upon the appearance of cyclin B mRNA in MCF10A cells (n=?). This pattern is similar as that in cancer cells and does not completely fit to the statement in lines 308-309 (different A3B expression patterns in cancer and non-cancer cells). Another study (PMID: 33592502) also showed that A3B mRNA were enriched in G2/M-phase in normal blood cells.

Response: Our statement “These findings indicate that expression of A3B may coordinate with the cell cycle in cancer cells but is unable to do so in normally proliferating non-cancerous tissue.”  is meant to confer the fact that non-cancerous tissues do not express A3B (probably as they are most often in the resting G0 phase of the cell cycle and are thus not stimulated), whereas in cancer cells and tissues, A3B, when capable of being expressed, is associated with proliferating cells (and peak levels during the G2 phase of the cell cycle). We also try to convey these key points in Figure 5. In response to this comment, to convey the message of this manuscript more clearly, we have made additional changes throughout the manuscript, including changing the title to “Aberrant APOBEC3B expression in breast cancer is linked to proliferation and cell cycle phase”.

Please also note that we have already referred to the paper by Hirabayashi et al. on APOBEC expression in myeloma as reference 22, and we have also included other references that have reported associations of APOBEC with proliferation.

  1. Considering the coordination of RNA and protein expression levels, whether A3B protein and/or mRNA modifications affect their amounts across various cell cycle phases and different cell lines?

Response: Our study does not address protein or RNA modifications, so we are regrettably unable to answer this interesting question.

Round 2

Reviewer 1 Report

The authors have answered most of the questions and have made suitable revisions.  However, there are no scale bar on pictures and no x-bar on some of columns.

Author Response

We thank the reviewer for this observation. We have now added scale bars to figure 1. For figure 2A, these data were N=1 from the CCLE database. To be able to add SD values to the bars, as we believe the reviewer requests, we have now performed a triplicate qPCR on available RNA.

Reviewer 2 Report

This reviewer acknowledges authors’ detail explanation to my questions and the revision of the manuscript, which is suitable for publication in the present form.

Author Response

we thank the reviewer for her/his time and effort in improving our manuscript.